# Risk to rely on soil carbon sequestration to offset global ruminant emissions

Yue Wang [1] ✉, Imke J. M. de Boer[1], U. Martin Persson [2], Raimon Ripoll-Bosch [1], Christel Cederberg [2], Pierre J. Gerber[1,3], Pete Smith [4] & Corina E. van Middelaar[1]

Carbon sequestration in grasslands has been proposed as an important means to offset greenhouse gas emissions from ruminant systems. To understand the potential and limitations of this strategy, we need to acknowledge that soil carbon sequestration is a time-limited benefit, and there are intrinsic differences between short- and long-lived greenhouse gases. Here, our analysis shows that one tonne of carbon sequestrated can offset radiative forcing of a continuous emission of 0.99 kg methane or 0.1 kg nitrous oxide per year over 100 years. About 135 gigatonnes of carbon is required to offset the continuous methane and nitrous oxide emissions from ruminant sector worldwide, nearly twice the current global carbon stock in managed grasslands. For various regions, grassland carbon stocks would need to increase by approximately 25% − 2,000%, indicating that solely relying on carbon sequestration in grasslands to offset warming effect of emissions from current ruminant systems is not feasible.

Climate change, resulting from a gradual increase in atmospheric greenhouse gas (GHG) concentrations, is one of our most pressing global challenges[1]. Our food systems are estimated to release about a third of all human-induced GHG emissions[2], with ruminant sector being a major source of anthropogenic methane ($CH_4$) and nitrous oxide ($N_2O$) emissions[3–6]. Given the urgency of mitigating GHG emissions from the global ruminant systems, a plethora of mitigation strategies have been suggested[7,8]. One such strategy is sequestering carbon (C) in soils on land that provides feed to the animals, i.e., net removal of carbon dioxide ($CO_2$) from the atmosphere[9–12]. Especially, since grasslands allocate higher proportions of biomass production belowground, they tend to hold higher soil organic carbon (SOC) stocks than croplands with predominantly annual crops[13]. Soil C-sequestration, however, is usually assumed to be temporary and it is a common perception that there is an upper limit to the amount of C that can be sequestered[14]. In many cases, sequestration rates decline to zero as the SOC stored reaches a new equilibrium. Without further disturbance, soils can become stable stores of C even if sequestration has stopped[15].

The cooling effect of soil C-sequestration is not well embedded in current GHG inventories[10]. To explore the potential for SOC increases to mitigate climate change, existing studies generally determine one single value, calculated as the sum of annual GHG emissions minus $CO_2$ removal by soil C-sequestration[11,16]. In certain circumstances, and based on short-term data only, it has been suggested that C-sequestration in grasslands could completely offset the emissions from the grazing system[11]. Solely focusing on the process of soil C-sequestration, however, makes the results largely depend on the C-sequestration stage of the soil (how far it is from reaching equilibrium) and fails to reflect that, in most cases, the potential soil C-sequestration is finite. Instead, translating cumulative soil C-sequestration into a final SOC stock change would allow its long-term impact in GHG mitigation to be better understood.

Furthermore, to sum the GHGs into one number, most studies express their climate impact in $CO_2$-equivalents ($CO_2$-eq) using the global warming potentials (GWPs). Equal $CO_2$-eq for different GHGs implies equal integrated radiative forcing of one emission pulse

[1]Animal Production Systems group, Wageningen University & Research, P.O. Box 338, 6700 AH Wageningen, the Netherlands. [2]Physical Resource Theory, Department of Space, Earth & Environment, Chalmers University of Technology, Gothenburg, Sweden. [3]The World Bank Group, 1818 H Street NW, Washington, DC 20433, USA. [4]Institute of Biological and Environmental Sciences, University of Aberdeen, 23 St Machar Drive, Aberdeen AB24 3UU, United Kingdom. ✉e-mail: yue3.wang@wur.nl

over a certain timeframe, but it says little about the contribution of the emission pulse of a gas to radiative forcing - and temperature change - at a certain point in time[17,18]. In other word, GWPs mask the end-point impact of the emissions and therefore, is considered inappropriate for the goals of the Paris Agreement[19]. Moreover, it does not account for temporal differences in climate impacts between short- and long-lived GHGs[20–22]. Comparing the impact behaviors of the same amount of $CH_4$ and $CO_2$, for example, $CH_4$ has a much higher impact on radiative forcing than $CO_2$ (i.e., approximate 120 times higher in the first year after the emission) and a much shorter perturbation lifetime (11.8 years for $CH_4$ and millennia for $CO_2$)[17,23]. This leads to markedly different impacts over the long term. The GWPs calculation with a 100-year time horizon, for instance, would suggest that if $CH_4$ emissions continue after year 100, additional soil C-sequestration would be needed to offset the warming from the additional emissions. This, however, is not the case since $CH_4$ is continuously broken down and removed from the atmosphere, therefore its climate effect stabilizes at a certain level after decades when emissions are constant[18].

Capturing this difference between long- and short-lived GHGs is precisely the logic behind the GWP*, which relates the climate impact of a one-off release of $CO_2$ to a change in the rate of $CH_4$ emissions[18,24]. However, GWP* has been criticized for its reliance on (arbitrary) baseline emissions (a grandfathering principle), resulting in unfair comparisons between countries in their contribution to warming[25,26]. In a situation where livestock numbers and associated $CH_4$ emissions are stable, GWP* of $CH_4$ is nearly zero (if not considering the delayed response of stock)[20]. However, although there is no additional warming under a constant level of $CH_4$ emissions, the historical emissions are still warming the planet (compared to what would have happened without those emissions) and maintaining ongoing damages from climate change[25]. Using a climate model allows to sidestep the arbitrary choice on baseline emissions while accounting for historical warming. Such a method, to our best knowledge, has rarely been used to incorporate soil C-sequestration in the GHG accounting of ruminant systems.

In this work, to improve the quantification of the GHG mitigation effect of soil C-sequestration in grasslands in ruminant systems, we introduce an alternative approach that fits into this particular purpose while overcoming the shortcomings of GWPs or GWP*. To this end, an existing climate model[27] was adopted to assess the (cumulative) climate impacts of GHGs fluxes over time, allowing the climatic differences between short-lived GHG emissions and (theoretically) long-lived but finite soil C-sequestration to be accounted for. The model was applied to estimate the required soil C-sequestration to cancel the $CH_4$ and $N_2O$ emissions from ruminant systems across the globe, which was found to be nearly double the current SOC stock in global managed grasslands. For various world regions, the current SOC stocks in managed grasslands need to be up to twentyfold to offset the regional emissions from ruminants. Those gaps provide an indication of how infeasible it is that soil C-sequestration in grasslands can truly cancel the warming effect of GHG emissions from ruminant systems.

## Results
### Climate impact of different GHGs
We first detail the climate impact of an emission pulse and a continues flow of emissions over time for each of the three gases (i.e., $CO_2$, $CH_4$, and $N_2O$), based on a climate model[27]. If we look at a pulse emission of one tonne (t) of $CO_2$, $CH_4$, and $N_2O$, we see clear differences in their lifetime and warming effect in terms of radiative forcing (RF; Fig. 1a). The RF of a pulse emission of one t of $CO_2$, for example, decreases from 0.0017 in year one to 0.0007 nW m$^{-2}$ in year 100, and 0.0005 nW m$^{-2}$ in year 500. Thus, the RF of $CO_2$ decays slowly and exists for centuries. $CH_4$ has a much higher impact on RF than $CO_2$. Meanwhile, the RF of one t of $CH_4$, as a result of its relative short perturbation lifetime, drops drastically within a few decades (i.e., from 0.2 nW m$^{-2}$ in year one to

0.0003 nW m$^{-2}$ in year 100 and 0.00005 nW m$^{-2}$ in year 500). $N_2O$ has a relative longer perturbation lifetime (109 years) and is also more potent compared to $CO_2$ and $CH_4$[23]. The RF associated with a pulse emission of one t of $N_2O$ gradually reduces over time and lasts for more than half a millennium (i.e., 0.36 nW m$^{-2}$ in year one and 0.005 nW m$^{-2}$ in year 500) (Fig. 1a).

When we examine the global surface temperature change associated with a pulse emission of GHGs (Fig. 1b), we see a pattern that largely follows the previously described RF curves. It is worth noting, though, that due to the inertia of the climate system it takes over 500 years for the temperature change from a pulse emission of $CO_2$ to exceed that of an equal pulse emission of $CH_4$, i.e., while the perturbation lifetime of $CH_4$ is short, its resulting climate impacts are far from so.

When it comes to a continuous flow of emissions, we see fundamental differences in RF or global temperature change between GHGs (Fig. 1c, d). The RF of a continuous flow of $CO_2$ emissions increases (practically) indefinitely, while the RF of a continuous flow of $CH_4$ and $N_2O$ stabilize with time. This is caused by the difference in lifetime between GHGs. Due to the long persistency of $CO_2$ in the atmosphere, parts of the $CO_2$ stay in the atmosphere (practically) indefinitely. A continuous flow of $CO_2$ emissions, therefore, causes a gradual increase in $CO_2$ concentration, and thus in radiative forcing and temperature change. Added $CH_4$ and $N_2O$ ultimately break down due to chemical processes and, therefore, are removed from the atmosphere (Fig. 1c). Consequently, a new equilibrium is stabilized where emissions and removals are approximately balanced. The longer the perturbation lifetime of the gas, the longer the time it takes to reach the new equilibrium (i.e., a few decades for $CH_4$ and more than 500 years for $N_2O$). When expressed in global surface temperature change, it takes even more years for the impact of continuous $CH_4$ and $N_2O$ emissions to find this new equilibrium but it still shows a trend towards stabilization, while continuous emissions of $CO_2$ add cumulatively to global temperature change (Fig. 1d).

### Equating a continuous emission of $CH_4$ or $N_2O$ with a pulse of $CO_2$
The above-described RF patterns show that the impact of a continues flow of $CH_4$ stabilizes while that of $N_2O$ shows the trend to stabilize (Fig. 1c). This offers the possibility to offset the impact of those two gases by a permanent C-sequestration, potentially offered by grassland soils. The continuous flow of $CO_2$ is not discussed here since its climate impact keeps accumulating and cannot be equated to a one-off sequestration. Moreover, it is believed that a fossil-free world is essential for climate change mitigation and (stringent) climate policy scenarios imply that agricultural energy use would eventually be largely decarbonized[28,29]. Soil C sinks can then be (potentially) used to offset the emissions from biological activities, for example, enteric $CH_4$ emissions.

Figure 2 shows how much $CO_2$ needs to be sequestrated to offset the climate impact of a continuous emission of one t of $CH_4$ or $N_2O$ across time. Given a 100-year timeframe, for example, the amount of $CO_2$ that needs to be stored to compensate the climate impact of a continuous flow of $CH_4$ (one t per year) equals about 3.7 kt for RF and 3.5 kt for global surface temperature change (Fig. 2). In other words, a one-off sequestration of one t of C would offset the RF of a continuous emission of 0.99 kg $CH_4$ per year or the global temperature change of 1.05 kg $CH_4$ per year over 100 years. These results are within the range of values found in earlier work equating continues emissions of $CH_4$ with a pulse emission of $CO_2$[30]. It is worth noting that while the climate benefits of $CO_2$ removal fade away, the climate impacts of $CH_4$ slightly increase (Fig. 1). This implies that more C needs to be sequestrated to offset continuous emissions over a longer timeframe (Fig. 2). This especially holds for the more persistent gas $N_2O$. The conversion ratio between, for example, the RF benefits of one pulse of $CO_2$

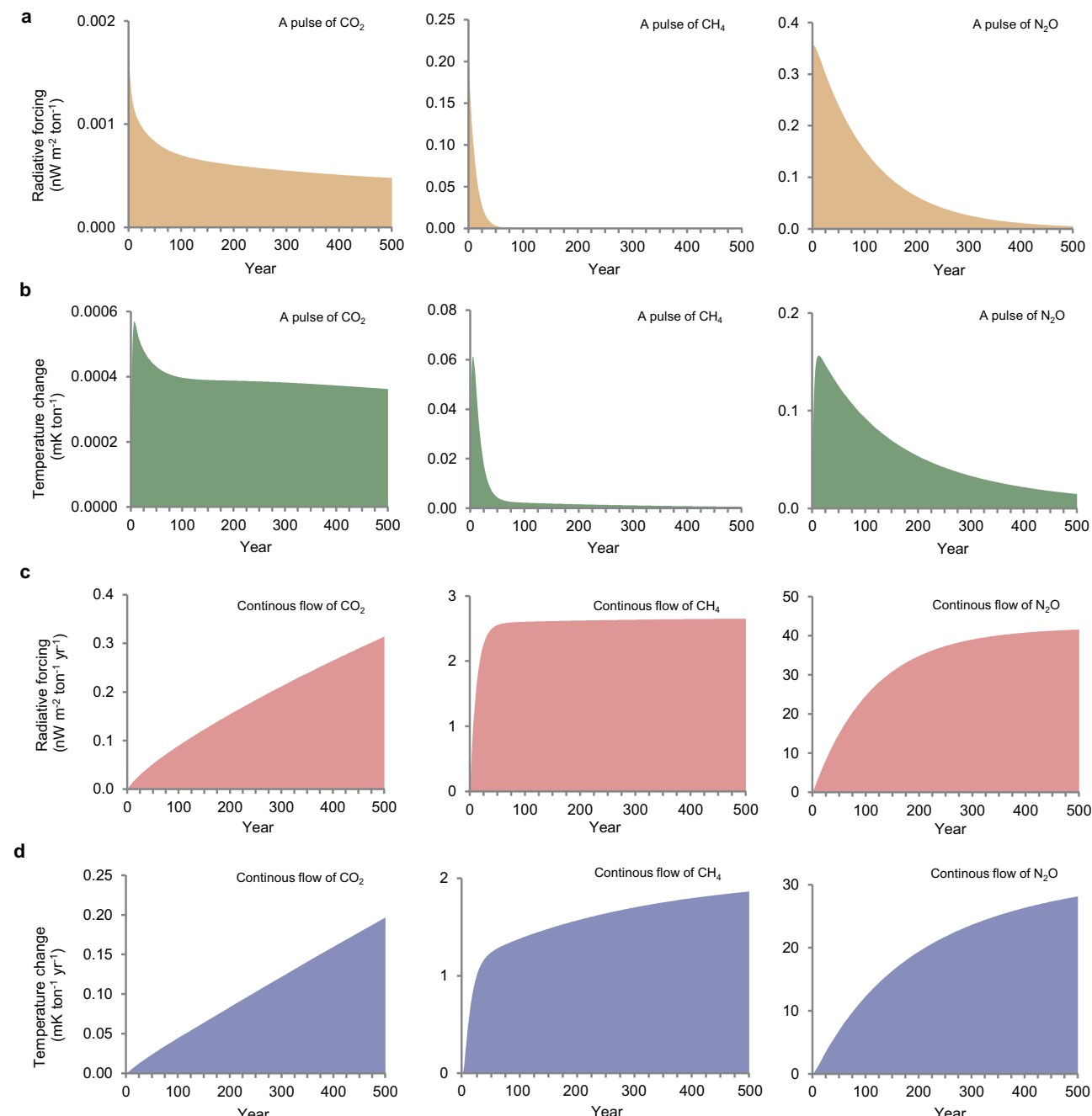

**Fig. 1 | Climate impact related to the emission of a greenhouse gas (GHG) including CO₂, CH₄ and N₂O. a** Radiative forcing of one pulse emission of a tonne (t) of GHGs during 500 years. The area under the graph represents the cumulative radiative forcing. **b** Global surface temperature change of one pulse emission of a t of GHGs during 500 years. **c** Cumulative radiative forcing of a continuous yearly emission of one t of GHGs during 500 years. **d** Cumulative global surface temperature change of a continuous yearly emission of one t of GHGs during 500 years. Note the difference in scale on the y-axis. The calculations are based on a simple climate model[27]. Source data are provided as a Source Data file.

sequestration and the RF impact of a continuous flow of N₂O emissions increases from 35 kt at year 100 to 87 kt at year 500 (Fig. 2). In other words, a one-off sequestration of one t of C would offset the RF of a continuous emission of 0.1 kg N₂O per year over 100 years or 0.04 kg N₂O per year over 500 years.

**An approach to embed soil C-sequestration in GHG calculations**
Cumulative soil C-sequestration (considered as a SOC stock increase) can be embedded in GHG accounting by applying the conversion ratios displayed in Fig. 2, as these equate the climate impact of a continuous flow of emissions of CH₄ or N₂O with a one-off pulse of

CO₂. In our approach, the long-term soil C-sequestration was translated into a one-off pulse of CO₂ in year one to overcome data limitations and to facilitate the application of the approach. To understand the potential implications of this practice, we tested three alternative scenarios to represent the long-term process of soil C-sequestration (Supplementary Note 1 and Supplementary Fig. 1). The results are largely consistent with our estimates, indicating that our assumption to translate C-sequestration into a one-off pulse of CO₂ is justifiable.

Accounting for soil C-sequestration is especially relevant for ruminant systems, which are dominant CH₄ emitters while being fed

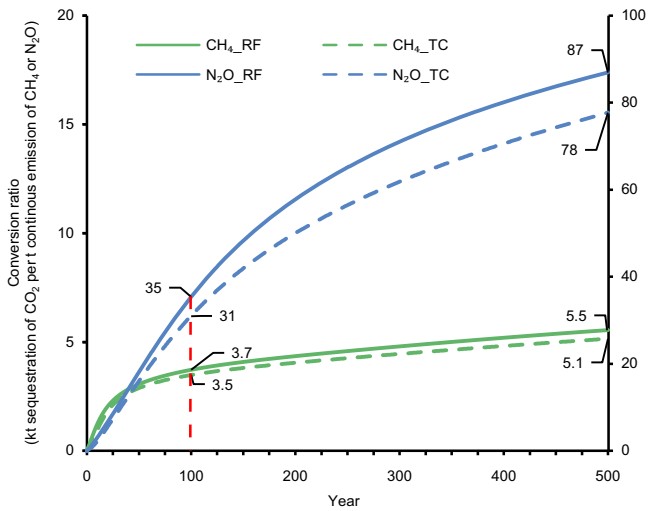

**Fig. 2 | The conversion ratios between the climate impact of a one-off pulse of $CO_2$ and a continuous emission of $CH_4$ or $N_2O$ over 500 years.** Climate impact is reflected via radiative forcing (RF) and global surface temperature change (TC). The values indicate how much $CO_2$ needs to be sequestrated to offset the climate impact of a continuous emission of one tonne (t) of $CH_4$ (main Y-axis) or $N_2O$ (secondary Y-axis). The red dashed line indicates the results at year 100. The calculations are based on a simple climate model[27]. Source data are provided as a Source Data file.

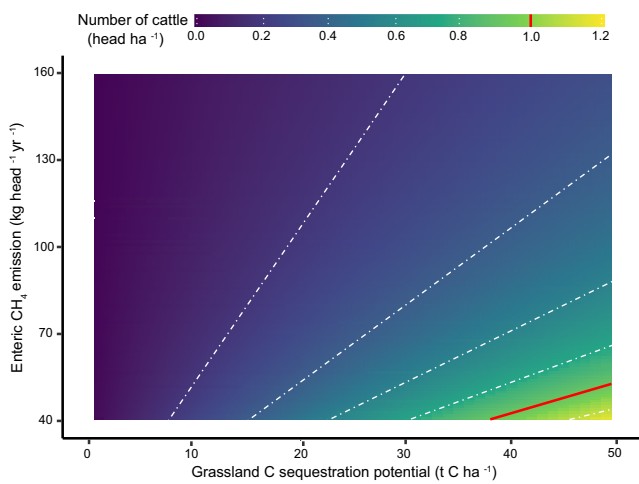

**Fig. 3 | Number of cattle 'allowed' for a given (maximum) soil carbon (C) sequestration in grasslands to offset the climate impact in a range of possible enteric $CH_4$ emissions per cattle.** It is assumed that one tonne (t) of C would offset a continuous emission of 0.99 kg $CH_4$ per year over 100 years, calculated based on a simple climate model[27]. The values for enteric $CH_4$ emission represent a wide range of cattle types, including productivity, animal age, breed, size, diet, production systems and so on[31]. The red line (in both chart area and legend) indicates a cattle density of one head per hectare of grassland. Source data are provided as a Source Data file.

on grasslands which represent an important store of SOC. Moreover, the approach is particularly valid for $CH_4$ (less time-varying) compared with $N_2O$. Therefore, we propose to first compare the climate benefit of soil C-sequestration with continuous $CH_4$ emissions from the ruminant sector. Only when the impact of continuous $CH_4$ emissions is completely offset by soil C-sequestration, we start comparing the impact of (residual) soil C-sequestration with continuous $N_2O$ emissions. The net balance of GHG emissions (i.e., emissions minus soil C-sequestration) might be summed based on commonly used metrics (e.g., GWPs or GWP*) to align with figures from existing studies. The following sections further illustrate how the conversion ratios we proposed could be applied into the context of ruminant systems. We hereafter adopt the conversion ratios of RF with a 100-year horizon for demonstration.

**Determining cattle density**
Our approach allows us to quantify the number of ruminants allowed on a given area of grassland with a specified capacity to store additional C, when aiming to offset the GHG emissions from the animals by soil C-sequestration. Here we take the cattle sector as an example to illustrate the concept focussing primarily on enteric $CH_4$ fermentation. The concept was also applied to manure $N_2O$ emissions (Supplementary Note 2) as well as the combination of enteric $CH_4$ and manure $N_2O$ emissions, which is displayed in Supplementary Figs. 2 and 3.

Figure 3 demonstrates how many cattle we could theoretically keep if aiming to completely offset enteric $CH_4$ emission over a 100-year period through soil C-sequestration. The range of values for enteric $CH_4$ emission (40–160 kg head$^{-1}$ year$^{-1}$) of cattle was derived from default values of Intergovernmental Panel on Climate Change (IPCC)[31], and represents a wide range of cattle types, including productivity, animal age, breed, size, diet, production systems and so on.

The (maximum) soil C-sequestration potential among grasslands (0-50 t ha$^{-1}$) was estimated based on the IPCC values, including default SOC stocks (at 30 cm depth) and the relative stock change factors for land management across different climate zones and soil types[31] (Methods). In the most optimistic case (i.e., maximum C-sequestration potential with least $CH_4$ emission), compensating a continuous annual

flow of 40 kg $CH_4$ per cattle would require an additional amount of nearly 40 t SOC to be stored in the grassland, which equals about 0.8 hectares of grassland representing a C-sequestration potential of 50 t ha$^{-1}$. In other words, one hectare of grassland potentially sequestering an additional 50 t SOC can compensate enteric $CH_4$ emissions of about 1.25 heads of cattle. The low values of cattle numbers in Fig. 3 indicate that soil C-sequestration potential in grasslands can only possibly cancel out a continuous flow of enteric $CH_4$ emissions in rather extensive systems (mostly with a cattle density lower than one head per hectare), whereas the density in practice is generally much higher than that[32,33]. It is worth emphasizing that, compared with the findings from other studies which investigated soil C-sequestration potential[34,35], a SOC increase of 50 t ha$^{-1}$ seems to be rather challenging and might be rarely reached. This, however, further strengthens our conclusions and concern that current animal densities are too high to fully compensate climate impacts by means of soil C-sequestration in grasslands.

**Soil C gaps for ruminant systems across the globe**
We further applied the conversion factors to estimate the required cumulative soil C-sequestration (i.e., required SOC stock increase) to offset $CH_4$ and $N_2O$ emissions from ruminant systems across the globe. Based on the Global Livestock Environmental Assessment Model (GLEAM 3.0)[36], the global ruminant sector (i.e., cattle, buffaloes, goats and sheep) released approximately 110 megatons of $CH_4$ and 2.4 megatons of $N_2O$ annually, with $CH_4$ emission from cattle as the dominant source. To offset these emissions, we need to increase SOC stock in global grassland by 135 gigatonnes (Gt) of C (Fig. 4a). This is almost equal to all SOC loss due to agriculture in the past 12000 years[37]. Note that the result was estimated using the simple linearized climate model with a main purpose of giving a sense for orders of magnitude (see Discussion for the uncertainty of the model).

Estimation of the extent of grasslands globally and their current SOC stocks vary across studies due to the definition of grasslands and data uncertainties[35,38,39]. Here, we based the extent of managed grasslands and their SOC stocks on the Global Soil Sequestration Potential Map[40], which results in an estimated global SOC stock of 78 Gt in

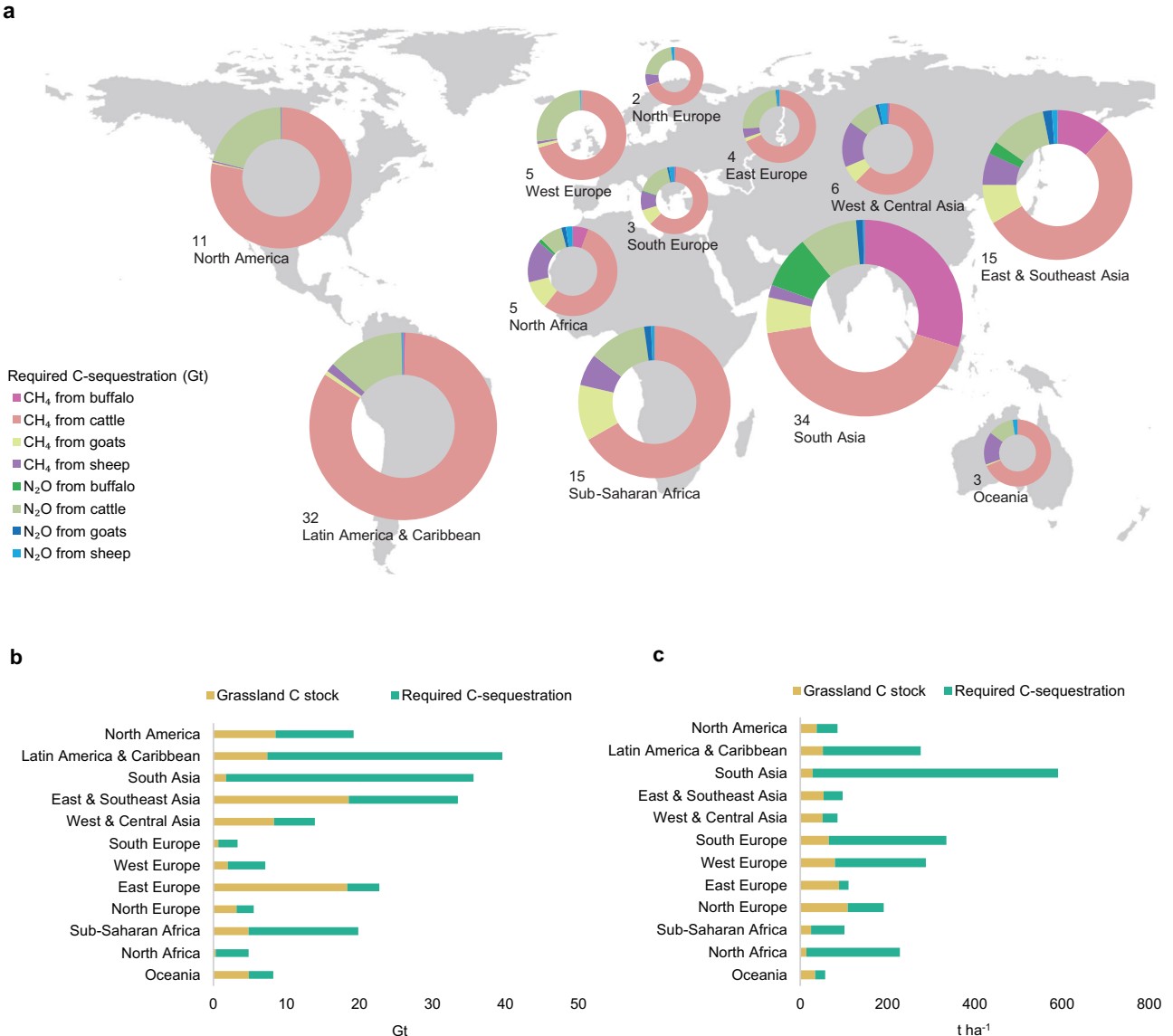

**Fig. 4 | Soil carbon (C) having gaps for ruminant systems across the globe. a** The required amount of C-sequestration to offset the continuous emissions of $CH_4$ and $N_2O$ from ruminants in different world regions over 100 years. The sizes of doughnut charts reflect the total amount (in gigatonnes, Gt) of required C-sequestration and the compositions of the doughnut charts explain the contributions of different ruminant species and different gases. The calculations were based on the Global Livestock Environmental Assessment Model (GLEAM 3.0)[36] and this study. The base map for global regions was derived from Esri datasets. **b** The required soil C-sequestration and current soil C stock in managed grasslands, in terms of total amount for each region. **c** The required soil C-sequestration and current soil C stock in managed grasslands, in terms of per hectare of land for each region. The data for soil C stock was derived from Global Soil Sequestration Potential Map (GSOCseq V1.1)[40].

grasslands used for ruminants (Fig. 4b). This means we would have to almost triple the current SOC stock in managed grasslands to offset current emissions. This is unlikely given the estimated C-sequestration potential in global grasslands of 0.6-2.2 Gt annually for a few decades[35,41].

The required SOC stock increase to offset $CH_4$ and $N_2O$ emissions from ruminant systems varies across global regions (Fig. 4a). However, for all regions, relying on soil C-sequestration in grasslands only is largely unfeasible, as SOC stocks would need to increase between about 25% and nearly 2,000%, depending on the region (Fig. 4b). For instance, South Asia has the highest total C-sequestration requirement (i.e., 34 Gt of SOC). This results in a requirement of 563 t SOC to be sequestered per ha of managed grassland, which is many times higher than the current SOC stock in grassland in the region (i.e., 28 t ha$^{-1}$) (Fig. 4c). Although it was reported that South Asia has relatively high potential for soil C-sequestration (about 6.6 t ha$^{-1}$) compared with

other world regions[35], clearly, solely relying on managed grasslands to offset the emissions will not suffice. Such large gaps were also found in many other regions, such as Latin America and the Caribbean, North America, South Europe, West Europe and African regions, in which the required C-sequestration amount is 1.3-15 times more than the current SOC stocks for per hectare of managed grassland (Fig. 4c). East Europe shows the smallest gap between current SOC stock and required C-sequestration in grasslands (Figs. 4b, c) but is reported to have a negligible sequestration potential in grassland soils[35]. As total SOC grassland stocks are particularly high in East and Southeast Asia and East Europe, effort should be put to prevent grassland degradation and/or conversion of grasslands into croplands to preserve current SOC stocks[15]. The differences in gaps between current stocks and required C-sequestration among regions are explained by multiple factors, such as differences in livestock density (animal number per hectare), extent of grasslands, farm management (e.g., level and type

of feed input, grassland management) or natural conditions (e.g., soil type and climate).

## Discussion

This study calls for critical thought regarding the potential of soil C-sequestration in grasslands to offset the climate warming caused by ruminant systems, in terms of both approach and implications. The approach we proposed offers important entry points for including soil C-sequestration in GHG calculations, which is especially relevant for those sectors which are large emitters of $CH_4$ and can act as SOC sink at the same time. Building on the approach, we demonstrate that when accounting for the differences between continued emission sources and sequestration of SOC, the claim that ruminant systems can have a negative annual GHG balance via soil C-sequestration is overly optimistic and could be misleading (at least at global or regional level).

To bridge the gaps between the amount of required soil C-sequestration and the current SOC stocks in grasslands, actions are needed in both reducing the sources of GHGs and in increasing SOC stocks. Reducing emissions is imperative. First, and as indicated in this study, reduction in $CO_2$ emissions and phasing out the use of fossil fuels is a must, since this has a persistent and permanent impact on the climate. To reduce the emissions of $CH_4$ and $N_2O$ from ruminant farming, several options have been proposed. We do not attempt to provide an exhaustive list of options here (see previous studies[7,8,42,43] instead), but those options encompass reducing livestock numbers; tackling feed availability and quality; improving animal health; inhibiting methanogenesis; and handling manure and urine depositions; in combination with other measures to reduce overall emissions. These options, however, need to be incorporated responsibly, considering the regional context and avoiding potential trade-offs (e.g., biodiversity loss, food-feed competition, livelihood of smallholders)[44–46]. Moreover, it has also been raised that promoting mobile pastoralism that mimics natural migratory herbivore systems could be a potential mitigation option when nature baselines were considered[47], yet it continues to be highly debated[48].

On the other hand, efforts should be put to restore degraded grasslands, preserve current C stocks in grasslands and further increase stocks where possible[35]. Here, again, a range of options are possible, and their potential largely depend on the regional context and the management of the grasslands. The continuum of intensity management options in grasslands will not only affect the capacity to increase SOC stocks in the soil, but also can lead to a wider range of benefits to society[49]. Furthermore, it is of equal importance to realize that SOC storage is one of the multiple functions of soils, among which trade-offs exist and need to be carefully addressed[50]. For instance, increasing SOC stocks does not necessarily result in an increase in yield and might lead to more $N_2O$ emissions[48,50].

Uncertainties and limitations exist in the study. Firstly, a 100-year timeframe has been chosen, which favors the positive impact of C-sequestration over the negative impact of GHG emissions, since the climate impact of a one-off removal of $CO_2$ continues decreasing over time while those of a continuous $CH_4$ and $N_2O$ emissions still increases after 100 years. Secondly, the yearly GHG emissions from ruminants were assumed to be constant as a simplified scenario, whereas global animal numbers are projected to increase rather than stabilize. The subsequent increase in emissions will further limit the capacity for soil C-sequestration to offset climate warming caused by ruminants. Thirdly, uncertainties in data and methodology exist when it comes to SOC stock and sequestration. For instance, the current SOC stock in those grasslands is highly uncertain and is sensitive to methodological choices (e.g., definition and extent of grassland, depth of soil, year of study)[35,51]. Soil C-sequestration is a dynamic process, and hence, SOC stocks are constantly changing, which raises concerns on the relevant timescales when crediting temporary C-sequestration[14]. The assumption on finiteness of soil C-sequestration is also disputed, which is

crucial in determining its long-term potential of C-sequestration[52]. Besides, by translating soil C-sequestration into a one-off pulse of $CO_2$ at year one, we neglected that it is a long-term process mainly due to the scarcity of well-established data that supports more reasonable alternative assumptions. Although our conclusions are not challenged by the exact timing and values of the soil C stock changes (Supplementary Fig. 1), better acquisition and accessibility to the data regarding variation at SOC stocks and potentials for C-sequestration in different soils and climates would be helpful for describing the role of soils in climate change mitigation in different circumstances[53]. Lastly, the analysis of this study was built on a simple linearized model, which was initially developed for small perturbations and may not necessarily reflect global aggregated values accurately as in our results. To get insight into the uncertainties of our global estimates, we tested the results using a reduced-complexity climate model (MAGICC7) that captures some of the bio-geophysical non-linearities in the climate system[54] (Supplementary Note 3). The outcome of this comparison (Supplementary Fig. 4) supports the main conclusion that it is not feasible to solely rely on soil C-sequestration in grasslands to offset warming effect of emissions from current ruminant systems.

In light of those limitations, this study does not attempt to precisely reveal to which extent C-sequestration can offset emissions from ruminant systems. Instead, we aim to demonstrate the principle of an approach to improve our understanding of the net climate impact of ruminant systems that act as a source and/or a sink for GHGs. In addition, we aim to raise awareness that under current systems, soil C-sequestration has a limited role to mitigate climate warming caused by the ruminant sector. This has important consequences for strategies to achieve climate neutrality in the livestock sector. Nonetheless, we acknowledge the importance of preserving and increasing current SOC stocks in grasslands, to mitigate climate change and to improve overall soil health for a range of other potential benefits (e.g., improve water retention, biodiversity or eventually grassland productivity).

## Methods

### Climate impact of different GHGs

We used a simple climate model based on the parametrization for calculating GWPs to demonstrate the climate impact of the emissions of different GHGs. It is a linearized representation of complex climate models, of which the underlying functions and values in the model have been introduced by Persson et al.[17] and updated by Persson & Johansson (2022) (Version 2.0)[27], based on the same set of assumptions regarding the radiative efficiency and perturbation lifetimes of the GHGs as used by IPCC AR6[23]. Carbon-cycle responses regarding $CH_4$ and $N_2O$ emissions are included compared with AR5, and the responses that measure indirect effects of changes in climate are also updated. Using these functions allows to evaluate the long-term climate impact of not only a pulse emission of GHGs, but also a continuous flow of the emissions.

The climate impact of GHGs is reflected in terms of both radiative forcing and global surface mean temperature change. The radiative forcing (expressed in nanowatt per square meter, $nW\ m^{-2}$) measures the energy input to the atmosphere-ocean system, which is affected by the GHG concentrations and is commonly used to indicate climate impact. Global surface temperature change (expressed in nanokelvin, nK) reflects the warming effect of the energy input in the average temperature of Earth surface, measured across land and ocean.

### Choice of time horizon

The climate model we used allows a choice of a flexible timeframe (up to a thousand years). Regarding the climate impact of different GHGs, we demonstrate the results for 500 years, which provides sufficient information to distinguish the differences between GHGs as well as revealing the long-term trend of the impact. To be consistent, when equating a one-off removal of $CO_2$ and a continuous emission of $CH_4$ or

$N_2O$, the conversion ratios over 500 years were displayed. While when demonstrating the concept and assess the effect of soil C-sequestration in GHG mitigation using the method we proposed, a timeframe of 100 years was chosen, given the reasons that 1) In most cases, the soils find the new equilibrium within 100 years and do not sequester C anymore unless changing the land use[15,31]; 2) The radiative forcing of continuous emission of $CH_4$ (the primary concerned gas in this study) stabilizes within 100 years; 3) Given the urgency to tackle global climate issues in this century, it is considered as more reasonable to focus on relatively near future than a much longer timeframe, which might be too far away to be realistically reachable; 4) A 100-year horizon is the most commonly-used timeframe in existing studies regarding GHG metrics, which allows our results to be compatible with other research.

### Emission factor of enteric $CH_4$ fermentation
IPCC default values (Tier 1) were used to derive enteric $CH_4$ emission factors of cattle (including dairy cattle and other cattle), which vary from 41-138 kg head$^{-1}$ year$^{-1}$, depending on region, age, breed, productivity and so on[30]. For dairy cattle, animal size and milk production are important determinants of emission, while for other cattle, animal size, population structure and production systems implemented are important determinants of emission factors. The lowest value (41 kg head$^{-1}$ year$^{-1}$) occurs in the Indian Subcontinent, who has the smallest cattle compared with other regions, whereas the highest value (138 kg head$^{-1}$ year$^{-1}$) occurs in highly productive commercialized dairy system in North America. For the purpose of display, we rounded the range of enteric $CH_4$ emission factor into 40-140 kg head$^{-1}$ year$^{-1}$.

### GHG emissions from global ruminant systems
The Global Livestock Environmental Assessment Model (GLEAM 3.0)[36] designed by Food and Agriculture Organization of the United Nations (FAO) was used to derive the data about total $CH_4$ and $N_2O$ emissions from the ruminant sector across the globe. The GLEAM model 3.0 assessed the GHG emissions of livestock supply chains in different regions using life cycle assessment, with 2015 as the reference year (latest data available). Ruminant animals that are included in the model are cattle, buffalo, sheep and goat. For each animal species, the emission of $CH_4$ (mainly from enteric fermentation and manure management) and $N_2O$ (mainly from feed production and manure management) were derived for 11 regions worldwide (i.e., North America, Latin America & Caribbean, South Asia, East & Southeast Asia, West & Central Asia, South Europe, West Europe, East Europe, North Europe, Sub-Saharan Africa, North Africa, Oceania). Emissions from feed production can originate from either the same or a different region where consumption takes place.

### Soil C stocks and sequestration potential in grasslands
The SOC stocks of managed grasslands (Fig. 4b, c) are derived from Global Soil Organic Carbon Sequestration Potential Map (GSOCseq V1.1) created by FAO[40]. The data demonstrates the current SOC stocks at mineral topsoil (0-30 cm, in t C ha$^{-1}$, -1 ×1 km resolution grid) in managed grasslands across the globe, with a reference year of 2020 (latest data available). The GSOCseq database was developed based on the submissions of FAO member countries (bottom-up approach), with the best available national layers on SOC stocks in t C ha$^{-1}$ for the first 30 cm of the topsoil layer and areas of their agricultural lands (including croplands and managed grasslands). When national layers on SOC stocks were not available, the reference values were retrieved by applying the methodology used for the GSOCseq v1.1. gap-fill countries, i.e., countries that could not submit a national map for the current version of the GSOCseq. These layers were generated following the same methodology for the GSOCseq v1.1. at a coarser resolution (of 5 km) using globally available data sources[40]. The output layers were subsequently downscaled to a 1 km resolution using a weighted

Generalized Additive Model following a boosted approach[55]. Furthermore, land cover from European Space Agency (http://www.esa-landcover-cci.org/) was used to derive the data of grasslands from the total agricultural lands for each country. Managed grassland here typically refers to the land that is dedicated to livestock production with a predominant herbaceous cover. More details about the map could be found via the FAO website https://www.fao.org/soils-portal/data-hub/soil-maps-and-databases/global-soil-organic-carbon-sequestration-potential-map-gsocseq/en/.

The soil C-sequestration potential of grasslands (Fig. 3) was estimated by comparing C stocks in poorly managed cropland and that in well managed grasslands. IPCC default values of C stocks in grasslands and croplands, as well as the relative stock change factors under different management practices, were used to calculate the differences in soil C stocks between poorly managed cropland (low input, full tillage) and well managed grasslands[31]. Such a difference was referred as the (maximum) soil C-sequestration potential of grasslands, given the fact that C stocks are typically higher in grasslands than in croplands. Among ten climate zones and six soil types defined by IPCC, six climate zones (i.e., tropical moist, tropical dry, warm temperate moist, warm temperate dry, cool temperate moist, cool temperate dry) and three soil types (i.e., high activity clay, low activity clay and sandy soil) were chosen in this study. The regions that are relatively rare at a global scale or have low cattle density were left out for the sake of representativeness[31,56–58]. Soil C-sequestration potential of grasslands in the selected regions vary from 5 to 50 t C ha$^{-1}$, with the lowest value occurs in sandy soils in tropical dry regions and the highest value occurs in high activity clay soils in cool temperate moist regions.

## Data availability

The climate model used in this study is available from Zenodo at https://zenodo.org/record/5957222. The data about methane emission factor from enteric fermentation and nitrous oxide emission factor from manure management of cattle are derived from the publicly available report of Intergovernmental Panel on Climate Change (IPCC), i.e., 2019 Refinement to the 2006 IPCC Guidelines for National Greenhouse Gas Inventories reports (https://www.ipcc-nggip.iges.or.jp/public/2019rf/index.html). The online tool of MAGICC7 could be found via https://live.magicc.org/. Global databases on soil carbon stocks and ruminant emissions are protected due to the data privacy laws and can only be available from Food and Agriculture Organization of the United Nations (FAO) on request. Source data are provided with this paper.

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

## Acknowledgements

We are sincerely grateful to Dominik Wisser (Animal Production and Health Division of Food and Agriculture Organization of the United Nations) and Isabel Luotto (Global Soil Partnership of Food and Agriculture Organization of the United Nations) for providing data and offering advice to this work. We also thank Sino-Dutch Dairy Development Center of China Agricultural University (to Y.W.) and the China Scholarship Council (No. 201906350227, to Y.W.) for the funding support. The views expressed in this document cannot be taken to reflect the official opinions of the supporting organizations.

## Author contributions

C.E.V.M. conceived and supervised the project. U.M.P. provided and examined the climate model. Y.W. and C.E.V.M. developed and implemented analyses. Y.W., C.E.V.M., I.J.M.B and R.R.B. wrote the manuscript, which was reviewed and edited by all of the authors.

## Competing interests

The authors declare no competing interests.
