## [Peer Review File · Nature Communications]

Reviewers' Comments:

Reviewer #1:

Remarks to the Author:

The authors suggested a new approach to assess the long-term climate impacts of ruminant systems that allows a (direct) comparison among the effects of different GHGs (CO₂, CH₄ and N₂O). The cumulative climate impacts of different GHG fluxes on radiative forcing and global mean surface temperature were derived from a simple climate model which is a linearized representation of complex climate models (like those in CMIP6). The authors further applied the approach to the ruminant systems that compared the climate effects of ruminant CH₄ and N₂O emissions and C-sequestration over grasslands. They found that based on the calculation, it is not feasible to fully offset the warming effects of ruminant emissions through increasing grassland carbon stocks. The analysis is of great importance to inform the potential of (grassland or even general) soil carbon sequestration in offsetting the increasing GHG emissions from ruminant systems (or other systems). The calculation of the soil C-sequestration vs. CH₄ emission, and the estimates of the requirement of SOC increase for mitigating ruminant CH₄ emissions are informative and reasonable. The manuscript is well written. But it might fit more a Perspective to raise this "shocking" knowledge with a different aspect of view (the new approach). My few major concerns are mainly for the "novel" approach used.

(1) The authors reviewed two sets of approach for comparing the effects of GHGs: the common-used matrix GWPs (used by IPCC reports) and the simple model GWP* for accounting the effects of short-life gases, and claimed "a novel approach that overcomes the shortcomings of GWPs or GWP*". However, it is not very clear/easy to understand what exactly is the novelty for understanding the climate effects. If people would like to know how much CO₂ needs to be sequestered to offset a CH₄ emission at 100-year timeframe, simply using the GWP would be enough. $1 \text{ kg CH}_4 = 29.8 \text{ kg CO}_2$, $1 \text{ kg CH}_4 * 100 \text{ year} = 2.98 \text{ ton CO}_2$ ($29.8 * 1 * 100$) = 0.81 ton C, which is very close to the value in this manuscript. The difference might even be due to the simplification of the climate model. The authors claimed to assess the effects on radiative forcing and global mean surface temperature (e.g., Fig. 1) while such effects can also be easily converted from CO₂ or CO₂e following historical or projected changes with simply multiplication. Given the concerns of climate change in near- and long-term, two GWPs at 20- and 100-year timeframe were given by IPCC report. It is not to say that GWPs is perfect, but it seems that it is good enough to obtain similar impacts and conclusion of this study (C-sequestration vs. CH₄ emissions at 100-year timeframe). Therefore, it is not clear why we should use a more complicated approach.

Furthermore, if one would like to know how much CO₂ needs to be sequestered to offset the effects of an increase in CH₄ emissions by e.g., 2015, GWP* model could be used. It is not clear how the approach in the manuscript is different from and superior to that of GWP*. GWP* infer the effects of increase/decrease CH₄ on climate (further warming or potential mitigation/cooling). It is indeed the historical emissions are still warming the planet, while the climate impacts of CH₄ decreases (mitigation) can be assessed too with GWP*. It is also not clear what is the advantage of this approach to GWP* for assessing climate impacts.

(2) The authors emphasized that they compared the effects of a continuous flow of emissions (e.g., CH₄) with a one-off pulse of CO₂. It is difficult to understand. For CH₄, it is understandable if we raise one cattle over 100-year emitting 40 kg CH₄ per year. However, for C-sequestration, it is not clear whether it means 40 ton C needs to be stored at the year 1 or 0.25 ton per year over 100-year? The finally effects at 2100 could be very different.

(3) As mentioned in the manuscript, the simple climate model is a linearized representation of more complex climate models. It is crucial to discuss how and to what extent this property will affect the numbers in this study. There are quite a few climate models like MAGGIC6 and ORSCAR that produce similar emission-climate impacts translations. How does the performance of this simple model to the others, whether the translation is correct, and what are the uncertainties?

Reviewer #2:

Remarks to the Author:

This paper does not pull its punches – wow, it gets to its critical point and hammers its message home. For reasons that escape many of us, carbon sequestration in grasslands has been proposed

as a means to offset greenhouse gas emissions from grazed cattle.. Many studies, based on no data, suggest that carbon will accumulate indefinitely and will offset ruminant methane emissions. The authors base their argument on the fact that soil carbon sequestration is a time-limited benefit and conduct a detailed analysis pointing out that there are intrinsic differences between short- and long-lived greenhouse gases.

Using a modification of an existing model, the authors calculate that grasslands would need to double C content in order to offset the radiative forcing of emitted ruminant gases. They introduce an approach to better account for the long-term climate impact of ruminant systems by accounting for the climatic differences between short-lived GHG emissions and long-lived C-sequestration. Their model is convincing and points out that that more C needs to be sequestered to offset continuous emissions of methane over a longer timeframe and that this especially holds for N₂O.

One of the most interesting calculations that they make, by assuming a maximum soil C-sequestration potential among grasslands based on the IPCC values, was that in order to compensate a continuous annual flow of 40 kg CH₄ per cattle, one hectare of grassland potentially sequestering an additional 50 t SOC can compensate enteric CH₄ emissions of about 1.25 heads of cattle and that cattle density is usually greater than that. This is great – but it does assume that grazing land can magically be managed to increase in soil C. This latter assumption is widely challenged, but that literature is not mentioned here or emphasized.

Overall this is a terrific paper; I just wish they would give a greater nod to the management-increasing-soil-C skeptics, which, frankly, is most academic soil scientists.

Reviewer #3:

Remarks to the Author:

This is an interesting and necessary paper that numerically describes the challenges of the much aired strategy of SOC fixation as a way to counterbalance GHG emission by ruminant livestock. The work is well conducted and the conclusions are solid, but the interpretation of the current state of the art both at the introduction and discussion make critical omissions that require the MS to be revised.

L 28-29 I would nuance this sentence, which currently reads too bluntly. Ruminants are indeed being attributed the largest share of anthropogenic CH₄, but increased evidence points to a non-negligible portion of such emissions being natural, and non-anthropogenic, in nature (see e.g., <https://doi.org/10.1038/s41612-023-00349-8>). The same perspective is completely ignored in the discussion in L.261-281, which is a critical failure of the paragraph. From the perspective that a non-negligible portion of livestock emissions belong to the ecosphere, the need to cut CH₄ emissions from the world systems with weakest inputs, or the need to improve their feed quality, becomes relative - to say the least.

L 36-39 This statement is again too bold and does not capture some work showing the possibility that soils are better at capturing C beyond the limits we had established (see e.g., <https://doi.org/10.1111/gcb.16804>). Worth commenting from ref 14 is the speed at which C is captured vs. the speed at which anthropogenic climate change is happening, which is often ignored.

Response to Reviewers

Reviewer #1

The authors suggested a new approach to assess the long-term climate impacts of ruminant systems that allows a (direct) comparison among the effects of different GHGs (CO₂, CH₄ and N₂O). The cumulative climate impacts of different GHG fluxes on radiative forcing and global mean surface temperature were derived from a simple climate model which is a linearized representation of complex climate models (like those in CMIP6). The authors further applied the approach to the ruminant systems that compared the climate effects of ruminant CH₄ and N₂O emissions and C-sequestration over grasslands. They found that based on the calculation, it is not feasible to fully offset the warming effects of ruminant emissions through increasing grassland carbon stocks. The analysis is of great importance to inform the potential of (grassland or even general) soil carbon sequestration in offsetting the increasing GHG emissions from ruminant systems (or other systems). The calculation of the soil C-sequestration vs. CH₄ emission, and the estimates of the requirement of SOC increase for mitigating ruminant CH₄ emissions are informative and reasonable. The manuscript is well written. But it might fit more a Perspective to raise this "shocking" knowledge with a different aspect of view (the new approach). My few major concerns are mainly for the "novel" approach used.

Response to Reviewer #1

Thanks very much for your time and acknowledgement on the importance of the analysis. Regarding your suggestion on transferring it into a Perspective paper, we believe that given the content of the paper, i.e., the originality of the work, the set-up and complexity of the method and the depth of the analysis, the study is more suited to be published as an article than as a perspective paper. However, we will further discuss this with the editor when appropriate. For your questions on the approach, please find our answers below and we hope that we have fully addressed your concerns.

Remark 1) The authors reviewed two sets of approach for comparing the effects of GHGs: the common-used matrix GWPs (used by IPCC reports) and the simple model GWP* for accounting the effects of short-life gases, and claimed "a novel approach that overcomes the shortcomings of GWPs or GWP*". However, it is not very clear/easy to understand what exactly is the novelty for understanding the climate effects. If people would like to know how much CO₂ needs to be sequestered to offset a CH₄ emission at 100-year timeframe, simply using the GWP would be enough. $1 \text{ kg CH}_4 = 29.8 \text{ kg CO}_2$, $1 \text{ kg CH}_4 * 100 \text{ year} = 2.98 \text{ ton CO}_2$ ($29.8 * 1 * 100$) = 0.81 ton C, which is very close to the value in this manuscript. The difference might even be due to the simplification of the climate model. The authors claimed to assess the effects on radiative forcing and global mean

surface temperature (e.g., Fig. 1) while such effects can also be easily converted from CO₂ or CO_{2e} following historical or projected changes with simply multiplication. Given the concerns of climate change in near- and long-term, two GWPs at 20- and 100-year timeframe were given by IPCC report. It is not to say that GWPs is perfect, but it seems that it is good enough to obtain similar impacts and conclusion of this study (C-sequestration vs. CH₄ emissions at 100-year timeframe). Therefore, it is not clear why we should use a more complicated approach.

Furthermore, if one would like to know how much CO₂ needs to be sequestered to offset the effects of an increase in CH₄ emissions by e.g., 2015, GWP* model could be used. It is not clear how the approach in the manuscript is different from and superior to that of GWP*. GWP* infer the effects of increase/decrease CH₄ on climate (further warming or potential mitigation/cooling). It is indeed the historical emissions are still warming the planet, while the climate impacts of CH₄ decreases (mitigation) can be assessed too with GWP*. It is also not clear what is the advantage of this approach to GWP* for assessing climate impacts.

Response 1) *Thanks for your valuable comments.*

Please allow us to first clarify that, the aim of this study is NOT to come up with an alternative approach to assess the climate impact of greenhouse gases, for which we do believe metrics like GWP and GWP are doing their job, despite of all the on-going debates. This paper, instead, focuses on how to incorporate soil C-sequestration in GHG accounting of ruminant systems, being one of the key issues causing current debates about the importance of C-sequestration as mitigation option. In this particular context, we propose our alternative approach, which is to equate the (cumulative) climate impact of continuous CH₄ and N₂O emissions with a one-off sequestration of CO₂. For this, we believe that the currently used GWP and GWP* metrics are less suited for the following reasons.*

First, taking the reviewer's example, using the AR6 GWP₁₀₀ for CH₄—which to be accurate is 27.2 t CO₂-eq /t CH₄—would suggest that sequestering 2,720 tons of CO₂ (3,500-3,700 tons in our study) would offset annual emissions of 1 ton of CH₄ over 100 years. However, comparing these two scenarios using the climate model from our paper suggests, sequestering 2,790 tons of CO₂ would reduce radiative forcing with 1.9 nW/m² or reduce global mean temperature with 1.08 nK at year 100, while the continuous emissions of 1 ton of CH₄ would increase radiative forcing with 2.6 nW/m² or increase global mean temperature with 1.38 nK - a difference of 27-38%.

While although this calculation gives an answer that is not totally off, numbers aside, the logic behind GWP and our method is fundamentally different. GWP is a time-integrated/cumulative metric, which compares the relative impacts of one pulse emission

integrated from the time of emission up to the chosen time horizon, e.g., 100 years. However, it doesn't say anything about the end-point impact at 100 years after the emission (see e.g., <https://doi.org/10.1016/J.ECOLIND.2016.06.049> for more explanations on the characteristics of GWP). This study focuses on the end-point impact of the emissions, which is believed to be better aligned with the temperature change at a certain point (see e.g., <https://doi.org/10.1088/1748-9326/ab6039>).

Second, the GWP calculation would suggest that if CH₄ emissions continued after year 100, additional carbon sequestration would be needed to offset the warming from the additional CH₄ emissions, but that is not the case. Because the CH₄ has a short atmospheric lifetime and is continuously broken down and removed from the atmosphere, continuous emissions will not add much warming after year 100. Instead, its cumulative climate effect stabilizes at a certain level (Fig 1c). This is also an intrinsic difference between CH₄ and CO₂. However, when calculating "1 kg CH₄ * 100 year = 2.98 ton CO₂ (29.8*1*100)" according to GWP, it assumes that the contribution of CH₄ emission from each year to climate is equal, which is not true. GWP fails to reveal the stabilization of the climate effect of the continuous CH₄ emissions (e.g., <https://www.nature.com/articles/nclimate2998>), and therefore, doesn't fit with the concept of our study. This is precisely the reason why we in this paper have chosen to use a climate model to better reflect the difference between long-lived and short-lived GHGs, as explained in detail in the "Climate impact of different GHGs" section of the manuscript (line 93-127).

For GWP*, yes, it is a smart approach to investigate the effects of an increase or decrease in CH₄ emissions on climate warming by looking at the rate change of emissions, and we did benefit from the concept of it. However, since it doesn't well address the warming from historical emissions (again, it emphasises on the rate change of emissions), under a constant flow, the climate impact using GWP* would be very minor. The fact that the historical emissions still warm the planet compared to what would have happened without those emissions leads us to arguing we cannot ignore this effect (e.g., see <https://iopscience.iop.org/article/10.1088/1748-9326/ac5930> for the criticism on GWP*). Compared with GWP*, a benefit of using the simple climate model is furthermore that, we do not need assumptions about baseline emissions (which can be contentious - see e.g. <https://doi.org/10.1088/1748-9326/ab4928>) - i.e., if doing the analysis using GWP*, it would make a huge difference for the assessment whether the livestock emissions we are comparing reflect constant, increasing or decreasing emissions (compared to some baseline). Using the climate model allows us to sidestep this contentious, and ultimately arbitrary issue.

For all those reasons, it is believed that our approach is more appropriate for the given context - incorporating soil C-sequestration in GHG accounting of ruminant systems. After subtracting the C-sequestration from total emissions using our approach, as indicated in line 186-188, the net GHG balance (residual emissions) might still be calculated based on the commonly used metrics (e.g., GWP or GWP).*

We hope we have clarified the matter. We have revised the corresponding lines in the Introduction and have better expanded the explanation of GWP and GWP to clarify the purpose of using a climate model as an alternative to already existing metrics (see below).*

Line 51-66: *"Furthermore, to sum the GHGs into one number, most studies express their climate impact in CO₂-equivalents (CO₂-eq) using global warming potentials (GWPs). Equal CO₂-eq for different GHGs implies equal integrated radiative forcing of one emission pulse over a certain timeframe, but it says little about the contribution of the emission pulse of a gas to radiative forcing - and temperature change - at a certain point in time ^{17,18}. In other word, GWP masks the end-point impact of the emissions and therefore, is considered inappropriate for the goals of the Paris Agreement ¹⁹. Moreover, it does not account for temporal differences in climate impacts between short- and long-lived GHGs ^{20,21, 22}. Comparing the impact behaviours of the same amount of CH₄ and CO₂, for example, CH₄ has a much higher impact on radiative forcing than CO₂ (i.e., approximate 120 times higher in year one) and a much shorter perturbation lifetime (11.8 years for CH₄ and millennia for CO₂) ^{17,23}. This leads to markedly different impacts over the long term. The GWP calculation would suggest that if CH₄ emissions continue after year 100, additional soil C- sequestration would be needed to offset the warming from the additional emissions. This, however, is not the case since CH₄ is continuously broken down and removed from the atmosphere, therefore its climate effect stabilizes at a certain level after decades when emissions are constant ¹⁸."*

Line 67-79: *"Capturing this difference between long- and short-lived GHGs is precisely the logic behind the GWP*, which relates the climate impact of a one-off release of CO₂ to a change in the rate of emissions of CH₄ ^{18,24}. However, GWP* has been criticized for its reliance on (arbitrary) baseline emissions (a grandfathering principle), resulting in unfair comparisons between countries in their contribution to warming ^{25,26}. In a situation where livestock numbers and associated CH₄ emissions are stable, GWP* of CH₄ is nearly zero (if not considering the delayed response of stock) ²⁰. However, although there is no additional warming under a constant level of CH₄ emissions, the historical emissions are still warming the planet (compared to what would have happened without those emissions) and maintain ongoing damages from climate change ²⁵. Using a climate model allows to sidestep the arbitrary choice on baseline emissions while accounting for*

historical warming. Such a method, to our best knowledge, has rarely been used to incorporate soil C-sequestration in the GHG accounting of ruminant systems.”

Line 80-82: *“ To improve the quantification of GHG mitigation effect of soil C-sequestration in grasslands in ruminant systems, this study introduces an alternative approach that fits into this particular purpose while overcoming the shortcomings of GWPs or GWP*..”*

Remark 2) The authors emphasized that they compared the effects of a continuous flow of emissions (e.g., CH₄) with a one-off pulse of CO₂. It is difficult to understand. For CH₄, it is understandable if we raise one cattle over 100-year emitting 40 kg CH₄ per year. However, for C-sequestration, it is not clear whether it means 40 ton C needs to be stored at the year 1 or 0.25 ton per year over 100-year? The finally effects at 2100 could be very different.

Response 2) *Thanks for pointing this out. For soil C-sequestration, we assumed that the C is stored at year 1. This means that we translate the long-term C-sequestration into a one-off removal of CO₂ at the very beginning, mainly because of lack of reliable data to incorporate the timeframe in the global assessment. However, while the reviewer is right that the timing of the C-sequestration does matter when talking about the soil process, we would like to emphasise it does not have a significant impact on the presented results. We made changes to the manuscript to overcome this comment but first let us explain the rationale.*

The graph below (referred as Supplementary Fig.1 now) shows the comparison in global temperature change between our assumption and some other alternative pathways (calculated using the same climate model as in the manuscript - calculations can be replicated using the model found here: <https://zenodo.org/record/5957222>). The four scenarios we compared are:

- 1) a 40 t C stock increase happening all in year 1 (this study);*
- 2) spread 40 t evenly over 100 years (i.e., 0.4 t C/yr, not 0.25 t C/yr), as indicated by the reviewer; This does lead to a different pathway as our assumption, whereas the final result at year 100 is—in relative terms—close.*
- 3) spread 40 t evenly over 20 years (i.e., 2 t C/yr). Because in most cases, soil C stocks due to management changes (e.g., grazing) are expected to equilibrate over periods shorter than 100 years, and a 20-year time perspective is widely used by e.g., IPCC guidelines and many other researchers. Switching to 20-year of C-sequestration implies that, although the paths within the first a few decades differ a bit, the differences between this assumption and our approach become negligible afterwards.*

4) a 10 t C stock increase in year 1 with an annual declining rate of 25% and a total sequestration amount of 40 t over 100 years (39 t over 20 years). To try to capture the fact that annual C-sequestration is not a constant value and it is likely to reduce over years, i.e., the closer to the new equilibrium, the less it can sequester, this trajectory is assumed to be the most representative of the 4 scenario's. The result from this scenario at year 100 is also very close to ours.

We recognize that reflecting the actual process of soil C-sequestration by incorporating the timeframe of that process would be the preferred option to show the expected climate impacts. The current status with regard to availability of sufficient and reliable data/evidence unfortunately prevents us from doing that at the global level. To our best understanding, the well-established global database on time-varying soil C-sequestration is still very scarce (due to e.g., the diversity between regions/contexts, the dynamic of soil process, difficulties to monitor soil in the long run). Thus, translating this long-term soil C-sequestration into a one-off pulse of CO₂ in year one is a conservative and defensible option, which also facilitates the application of the approach.

In conclusion, given the fact that our result is largely comparable with other assumptions (especially at year 100 and hereafter), and the main point with the analysis is whether soil C-sequestration in grasslands is able to offset ruminant emissions in the long-term, which is not sensitive to the exact timing of the soil C stock changes, we can show that our results are valid.

Supplementary Fig.1. Global temperature change over 200 years resulted by (a total amount of 40 t) soil carbon sequestration under different assumptions on the timing of sequestration. The calculations were conducted based on a simple climate model (Persson & Johansson, 2022).

We have further clarified this issue in the revised manuscript as follows, and have incorporated the four scenarios as well as the figure in the Supplementary Note 1:

Line 174-179: *"In our approach, the long-term soil C-sequestration was translated into a one-off pulse of CO₂ in year one to overcome data limitations and to facilitate the application of the approach. To understand the potential implications, we tested three alternative scenarios to represent the long-term process of soil C-sequestration (see Supplementary Fig.1). The results are largely consistent with our estimates, indicating that our assumption to translate C-sequestration into a one-off pulse of CO₂ is justifiable".*

Line 322-329: *"Besides, by translating soil C-sequestration into a one-off pulse of CO₂ at year one, we neglected that it is a long-term process mainly due to the scarcity of well-established data that supports more reasonable alternative assumptions. Although our conclusions are not challenged by the exact timing and values of the soil C stock changes (see Supplementary Fig.1), better acquisition and accessibility to the data regarding variation at SOC stocks and potentials for C-sequestration in different soils and climates would be helpful for describing the role of soils in climate change mitigation in different circumstances ⁵³."*

Remark 3) As mentioned in the manuscript, the simple climate model is a linearized representation of more complex climate models. It is crucial to discuss how and to what extent this property will affect the numbers in this study. There are quite a few climate models like MAGGIC6 and ORSCAR that produce similar emission-climate impacts translations. How does the performance of this simple model to the others, whether the translation is correct, and what are the uncertainties?

Response 3) *Thanks for this question. Indeed, the property of the model would have an impact on the aggregate numbers in our results. We have, therefore, tested our results using another climate model – MAGICC (<https://live.magicc.org/>). A middle-of-the-road baseline (SSP2-4.5) was selected as demonstration, where trends broadly follow their historical patterns and has a nominal 4.5 W m⁻² radiative forcing level by 2100. Based on our result "to offset the continuous emission of 110 Mt CH₄ and 2.4 Mt N₂O annually over 100 years, 135 Gt of C needs to be sequestered", we run the following scenarios built on SSP2-4.5: 1) a sequestration of 135 Gt of C (495 Gt of CO₂) from year 2020 to 2040; 2) avoid annual emissions of 110 Mt CH₄ and 2.4 Mt N₂O from year 2020 to 2100.*

The resulting impacts on radiative forcing and global mean temperature change can be seen in the graphs below (referred as Supplementary Fig.4 now). The temporal paths of the two scenarios diverge a bit, with the impact of carbon sequestration being larger in the beginning and the effect of avoided ruminant emissions being larger towards the end of the century (compared with baseline). This divergence reflects the difference between

the climate impacts calculated with the simple and the more complex model, and we believe that the difference would not challenge the main insights from our analysis, given that the soil carbon gaps are so big (see figure 4b). To implement this, the following changes have been made in the revised manuscript:

1) Results (one sentence added)

Line 237-239: "Note that the result was estimated using the simple linearized climate model with a main purpose of giving a sense for orders of magnitude (see Discussion for the uncertainty of the model)".

2) Discussion (a few sentences added)

Line 329-336: "Lastly, the analysis of this study was built on a simple linearized model, which was initially developed for small perturbations and may not necessarily reflect globally aggregated values accurately as in our results. To get insight into the uncertainties, we tested our global results using a reduced-complexity climate model (MAGICC7) that captures some of the bio-geophysical non-linearities in the climate system⁵⁴. The outcome of this comparison (see Supplementary Fig.4) supports the main conclusion that it is not feasible to solely rely on soil C-sequestration in grasslands to offset warming effect of emissions from current ruminant systems".

3) Supplementary Information (the figure as well as its methodological details added)

Supplementary Fig. 4. Climate impacts of two scenarios using MAGICC climate model. The scenarios were developed based on the global aggregated results of this study, with scenario 1 assumed a sequestration of 135 Gt of C from year 2020 to 2040, and scenario 2 assumed that annual emission of 110 Mt CH₄ and 2.4 Mt N₂O from year 2020 to 2100 were avoided. The scenarios were run and compared by the Model for the Assessment of Greenhouse Gas Induced Climate Change (MAGICC), and the shared socioeconomic pathways (SSP2-4.5), a middle-of-the-road scenario, was selected as a baseline.

Reviewer #2

This paper does not pull its punches – wow, it gets to its critical point and hammers its message home. For reasons that escape many of us, carbon sequestration in grasslands has been proposed as a means to offset greenhouse gas emissions from grazed cattle. Many studies, based on no data, suggest that carbon will accumulate indefinitely and will offset ruminant methane emissions. The authors base their argument on the fact that soil carbon sequestration is a time-limited benefit and conduct a detailed analysis pointing out that there are intrinsic differences between short- and long-lived greenhouse gases.

Using a modification of an existing model, the authors calculate that grasslands would need to double C content in order to offset the radiative forcing of emitted ruminant gases. They introduce an approach to better account for the long-term climate impact of ruminant systems by accounting for the climatic differences between short-lived GHG emissions and long-lived C-sequestration. Their model is convincing and points out that that more C needs to be sequestered to offset continuous emissions of methane over a longer timeframe and that this especially holds for N₂O.

One of the most interesting calculations that they make, by assuming a maximum soil C-sequestration potential among grasslands based on the IPCC values, was that in order to compensate a continuous annual flow of 40 kg CH₄ per cattle, one hectare of grassland potentially sequestering an additional 50 t SOC can compensate enteric CH₄ emissions of about 1.25 heads of cattle and that cattle density is usually greater than that. This is great – but it does assume that grazing land can magically be managed to increase in soil C. This latter assumption is widely challenged, but that literature is not mentioned here or emphasized.

Overall this is a terrific paper; I just wish they would give a greater nod to the management-increasing-soil-C skeptics, which, frankly, is most academic soil scientists.

Response to Reviewer #2

Many thanks for your compliments on this work! Regarding your concern, yes, we did assume that the soil could sequester 50 t C/ha calculated IPCC values (details have been given in Methods line 559-573). As we stated in the manuscript, this is considered as the most optimistic case, i.e., maximum C-sequestration potential (50 t C/ha) with least CH₄ emission (40 kg CH₄/cattle).

In our calculations, the difference in SOC between poorly managed cropland (low input, full tillage) and well managed grasslands in the regions was assumed to be the maximum C-sequestration potential (since grasslands are normally believed to have a higher C

stock than croplands), and this most optimistic value (50 t C/ha) occurs in high activity clay soils in cool temperate moist regions. We admit that this high value of C stock increase could indeed be challenging (e.g., see also the latest FAO report (<https://www.fao.org/3/cc3981en/cc3981en.pdf>), which is the first global assessment of soil carbon in grasslands reporting a maximum of 0.41 t C/ha/year to be achieved after 20 years of the application of management practices).

Whereas please let us emphasize that, for this section we do not aim to accurately show the numbers of soil C-sequestration potential in practice, but instead, to illustrate that - even under the most 'extreme' optimistic case, soil C-sequestration is unlikely to offset animal emissions. We believe that the difficulties of reaching "50t/ha" strengthens this conclusion and have added a few lines to the corresponding section to further clarify this point:

Line 217-221: "It is worth emphasizing that, compared with the findings from other studies which investigated soil C-sequestration potential ^{34,35}, an SOC increase of 50 t ha⁻¹ seems to be rather challenging and might be rarely reached. This, however, further strengthens our conclusions and concern that current animal densities are too high to fully compensate climate impacts by means of C-sequestration."

Reviewer #3

This is an interesting and necessary paper that numerically describes the challenges of the much aired strategy of SOC fixation as a way to counterbalance GHG emission by ruminant livestock. The work is well conducted and the conclusions are solid, but the interpretation of the current state of the art both at the introduction and discussion make critical omissions that require the MS to be revised.

Response to Reviewer #3

Thanks a lot for your acknowledgement and concerns! Please find our answers to your questions below. We appreciate your insightful comments and have incorporated them in the revised manuscript as well as we can.

Remark 1) L 28-29 I would nuance this sentence, which currently reads too bluntly. Ruminants are indeed being attributed the largest share of anthropogenic CH₄, but increased evidence points to a non-negligible portion of such emissions being natural, and non-anthropogenic, in nature (see e.g., <https://doi.org/10.1038/s41612-023-00349-8>). The same perspective is completely ignored in the discussion in L.261-281, which is a critical failure of the paragraph. From the perspective that a non-negligible portion of livestock emissions belong to the ecosphere, the need to cut CH₄ emissions from the world systems with weakest inputs, or the need to improve their feed quality, becomes relative - to say the least.

Response 1) *Thanks for your suggestion. The publication of Manzano et al. (2023) shows that there is a debate on CH₄ attribution in ruminant systems, with an argument that a portion of the emissions could be considered as "natural" if the animals are (partly) based on natural grassland. We also noticed that different voices exist on this issue (e.g., see page 80-82 in the report https://www.oxfordmartin.ox.ac.uk/downloads/reports/fcrn_gnc_report.pdf). We have incorporated this perspective into the Discussion of the updated manuscript, and have revised the sentence in the Introduction to avoid too blunt expressions (see below). We hope that in this way we have addressed your concerns. Thanks!*

Line 27-30: *"Our food systems are estimated to release about a third of all human-induced GHG emissions ², with ruminant sector being a major source of anthropogenic methane (CH₄) and nitrous oxide (N₂O) emissions ^{3,4,5,6}".*

Line 296-298: *"Moreover, mobile pastoralism that mimics natural migratory herbivore systems has been raised as a potential mitigation option if natural baselines are considered ⁴⁷, yet it continues to be highly debated ⁵⁰".*

Remark 2) L36-39 This statement is again too bold and does not capture some work showing the possibility that soils are better at capturing C beyond the limits we had established (see e.g., <https://doi.org/10.1111/gcb.16804>). Worth commenting from ref 14 is the speed at which C is captured vs. the speed at which anthropogenic climate change is happening, which is often ignored.

Response 2) Thanks for pointing this out. Objectively speaking, it is a common perception that there is a limit of soil carbon storage, which has been recently challenged by Begill et al. (2023). More recently, a letter (Cotrufo et al., 2023, <https://onlinelibrary.wiley.com/doi/10.1111/gcb.16921>) has been published to challenge the conclusion of Begill et al. (2023), and the argue is currently going on. As for the speed or timeframe of C-sequestration, we would like to refer to our response to remark 2 of Reviewer 1, where we explain that we incorporated three alternative scenarios (based on different assumptions of the timing of C-sequestration) to verify our approach.

In addition to this, the following revisions have been made based on your suggestions:

*Introduction, **line 35-40**: "Soil C-sequestration, however, is usually claimed to be temporary and finite, so it is a common perception that there is an upper limit to the amount of C that can be sequestered¹⁴. In many cases, sequestration rates decline to zero as the SOC stored reaches a new equilibrium. Without further disturbance, soils can become stable stores of C even if sequestration has stopped¹⁵."*

*Discussion, **line 318-322**:" Soil C-sequestration is a dynamic process, and hence, SOC stocks are constantly changing, which raises concerns on the relevant timescales when crediting temporary C-sequestration¹⁴. The assumption on finiteness of soil C-sequestration is also disputed, which is crucial in determining its long-term potential of C-sequestration⁵²."*

Reviewers' Comments:

Reviewer #1:

Remarks to the Author:

The authors very well addressed all my concerns. The revision is improved with additional clarifications and analysis. I believe it is a very nice piece of work raising an important but sometimes neglected issue on C sink capacities vs. GHG emissions.